# Real-World Outcomes in *FLT3*-ITD Mutated Acute Myeloid Leukemia: Impact of NPM1 Mutations and Allogeneic Transplantation in a Retrospective Unicentric Cohort

**DOI:** 10.3390/jcm14145110

**Published:** 2025-07-18

**Authors:** Veronica Vecchio, Andrea Duminuco, Salvatore Leotta, Elisa Mauro, Cinzia Maugeri, Marina Parisi, Paolo Fabio Fiumara, Francesco Di Raimondo, Giuseppe A. Palumbo, Lucia Gozzo, Fanny Erika Palumbo, Calogero Vetro

**Affiliations:** 1Hematology Unit with BMT, A.O.U. Policlinico “G. Rodolico-San Marco”, 95123 Catania, Italy; vero.vecchio99@gmail.com (V.V.); leotta3@yahoo.it (S.L.); elixmauro17@gmail.com (E.M.); maugericinzia@hotmail.com (C.M.); marinaparisi@hotmail.it (M.P.); paolo.fiumara@virgilio.it (P.F.F.); diraimon@unict.it (F.D.R.); palumbo.ga@gmail.com (G.A.P.); fannypalumbo@gmail.com (F.E.P.); 2Department of Medical, Surgical Sciences and Advanced Technologies “G.F. Ingrassia”, University of Catania, 95123 Catania, Italy; 3Clinical Pharmacology Program/Regional Pharmacovigilance Centre, A.O.U. Policlinico “G. Rodolico-San Marco”, 95123 Catania, Italy; luciagozzo86@icloud.com; 4Hematology and Bone Marrow Transplantation Unit (BMTU), Hospital of Bolzano (SABES-ASDAA), Teaching Hospital of Paracelsus Medical University (PMU), 39100 Bolzano, Italy; gerovetro@gmail.com

**Keywords:** acute myeloid leukemia, *FLT3*-ITD, midostaurin, allelic ratio, overall survival, ELN guidelines

## Abstract

**Background/Objectives**: Acute myeloid leukemia (AML) with *FLT3* internal tandem duplication (*FLT3*-ITD) mutations carries a poor prognosis. While *FLT3* inhibitors like midostaurin show benefits in combination with chemotherapy, the role of allelic ratio (AR), *NPM1* mutation status, and hematopoietic stem cell transplantation (HSCT) remains uncertain. Real-world data can help refine prognostic classification and treatment strategies. **Methods**: We retrospectively analyzed 37 fit patients with *FLT3*-ITD AML treated with standard “7+3” chemotherapy, with and without midostaurin, between 2013 and 2022. Patients were stratified by *FLT3*-ITD AR, *NPM1* status, and treatment approach. Outcomes assessed included complete remission (CR), disease-free survival (DFS), and overall survival (OS). **Results**: Overall, 67.6% achieved CR/CRi. Response rates did not differ significantly by AR (low vs. high: 66.7% vs. 69.2%) or midostaurin use (72.6% vs. 60%; *p* = 0.49). *NPM1* mutations were associated with improved DFS (10.3 vs. 3 months, *p* = 0.036) but not OS. HSCT, performed in 54.1% of patients, mainly in first remission (CR1), significantly prolonged DFS (not reached vs. 5.3 months, *p* = 0.005) and remained an independent predictor in multivariate analysis (HR: 0.160, *p* = 0.039). OS (median 15.1 months) did not vary significantly across subgroups. Among patients achieving CR1, OS was significantly longer in those who underwent HSCT after midostaurin-based induction compared to those not transplanted (median OS not reached vs. 12.8 months; 95% CI, 6.9–18.7; *p* = 0.045), whereas no significant benefit was observed after standard induction. In a landmark analysis restricted to patients transplanted in CR1, those who had received midostaurin-based induction showed a trend toward improved OS compared to those treated with standard induction (median OS not reached vs. 11.5 months; 95% CI, 0.5–25.0; *p* = 0.086). **Conclusions**: This real-life study supports the importance of *NPM1* mutations and HSCT in CR1, especially in the midostaurin era, for improving DFS in *FLT3*-ITD AML. These findings support updated guidelines for reducing the prognostic weight of AR and highlight the need for improved post-remission strategies in this setting.

## 1. Introduction

FMS-like tyrosine kinase 3 (*FLT3*) mutation is reported in 30% of acute myeloid leukemia (AML) cases, with the internal tandem duplication (ITD) representing the most common type of *FLT3* mutation (*FLT3*-ITD; approximately 25% of all AML cases) [1]. Patients with *FLT3-ITD* mutation have a poorer prognosis compared to those without, showing similar response rates to chemotherapy but with a higher risk of relapse, even if allogeneic hematopoietic stem cell transplantation (HSCT) can improve the outcome [2]. Indeed, *FLT3*-ITD mutated AML can relapse even as a hematological emergency, often presenting with an extremely high white blood cell (WBC) count and intracranial hemorrhages, complicating management and requiring urgent care [3,4,5].

In addition to ITD mutations, *FLT3* can also harbor point mutations. While *FLT3*-ITD is a well-recognized poor prognostic factor in AML, significantly diminishing both disease-free survival (DFS) and overall survival (OS), the *FLT3*-TKD (Tyrosine Kinase Domain) point mutation has demonstrated an inconsistent prognostic impact [6,7,8,9].

The proportion of mutated clones within neoplastic cells could in turn influence the prognostic impact of the ITD mutation. This is quantified through the allelic ratio (AR), which represents the number of ITD-mutated alleles relative to wild-type alleles. Determination of this ratio is achieved via DNA fragment analysis, utilizing polymerase chain reaction (PCR)-based methods in conjunction with capillary electrophoresis to ensure precise measurements [10]. A higher *FLT3*-ITD AR (≥0.5) was previously associated with worse survival than a lower ratio [11,12].

In previous years, the 2017 European LeukemiaNet (ELN) classification incorporated the AR as a relevant prognostic factor, stratifying patients according to *FLT3*-ITD AR and *Nucleophosmin 1* (*NPM1*) mutational status [12]. However, in the most recent 2022 ELN guidelines, its significance has diminished, and it is no longer considered a determining parameter in AML risk stratification [13]. While the AR may still provide some insight into disease biology, current evidence suggests that its prognostic relevance is limited compared to other established markers [10,12,13,14].

In *FLT3*-mutated AML treatment, induction therapy for fit patients typically involves intensive chemotherapy (as the “7+3” regimen) combined with *FLT3* inhibitors, such as midostaurin. *FLT3* inhibitors are divided into first-generation (e.g., midostaurin, sorafenib) and second-generation (e.g., quizartinib, gilteritinib) agents. First-generation inhibitors target both *FLT3*-ITD and *FLT3*-TKD, though their broader kinase activity extends beyond *FLT3*. Second-generation inhibitors, on the other hand, are more selective for *FLT3*-ITD with better safety profiles [8,15,16].

The RATIFY trial demonstrated that in patients aged 18 to 59 years with *FLT3*-mutated AML, adding midostaurin to standard chemotherapy significantly improved survival outcomes. The study found that this combination reduced the risk of death by 22% compared to chemotherapy alone (placebo) [17]. Patient randomization was stratified according to *FLT3*-ITD AR using a cutoff of 0.7, distinguishing between low (0.05–0.7) and high (>0.7) AR [17]. This threshold was derived from earlier studies suggesting a dose-dependent adverse effect of mutant burden [18]. However, subsequent analyses and independent cohort studies indicated that a cutoff of 0.5 provides a better prognostic separation between risk groups [10,19,20]. As a result, the 2017 ELN guidelines adopted 0.5 as the official cutoff, defining low AR as <0.5 and high AR as ≥0.5 [12]. The ELN classification further integrated *NPM1* mutation status, categorizing patients with *NPM1*-mutated/*FLT3*-ITD low AR as favorable risk, and those with *NPM1* wild-type or high AR as intermediate or adverse risk. This refined classification aimed to improve risk-adapted treatment strategies [13,17].

Recommendations for HSCT remain controversial. The former European Society for Blood and Marrow Transplantation (EBMT) recommendations suggested that all *FLT3*-ITD patients should undergo HSCT, with a grey zone represented by patients with *NPM1* mutations and low AR *FLT3*-ITD [21]. For them, HSCT could have been withheld in case of MRD negativity, as suggested by the ELN group [12], or pursued independently from MRD, as NCCN guidelines recommend [21,22]. However, with the recently revised classification emphasizing the intermediate-risk nature of *FLT3*-ITD in general, new considerations and guidelines need to be implemented. In this context, the last EBMT study reported a better outcome in the last years for transplanted *FLT3*-ITD patients without concomitant *NPM1* mutation, while this advantage over time was not seen in patients with concomitant *NPM1* mutation [23].

Despite advances in therapy, outcomes for patients with *FLT3*-ITD mutated AML remain suboptimal. Furthermore, the prognostic impact of AR and midostaurin requires further clarification, as their precise influence on disease progression is not fully understood. Additional studies are essential to refine treatment strategies and enhance the overall management of *FLT3*-mutated AML. For this reason, we aimed to assess the prognostic impact of *FLT3*-ITD mutations in fit AML patients who underwent intensive chemotherapy, both with and without midostaurin, at our center. By analyzing outcomes in this cohort, we sought to better understand how *FLT3*-ITD AR influences disease progression and treatment response, particularly in patients receiving targeted therapy.

## 2. Materials and Methods

### 2.1. Aim and Study Design

This retrospective, monocentric study analyzed the outcome of *FLT3*-mutated AML treated with the “7+3” regimen, with and without midostaurin, between 2013 and 2022. The prognostic assessment included response rates after induction therapy and survival outcomes, measured as DFS and OS. Patients diagnosed with *FLT3*-ITD-positive AML were included in the study if deemed fit for intensive chemotherapy according to SIE/SIES/GITMO criteria [24].

Only patients considered eligible for intensive chemotherapy according to SIE/SIES/GITMO criteria were included [24]. Patients with prior AML treatment, concurrent malignancies, or deemed unfit for induction therapy were excluded.

Before therapy, all patients underwent comprehensive diagnostic evaluations, including bone marrow aspirate and/or biopsy, cytogenetic and molecular analyses, complete blood count, liver and renal function tests, and cardiological assessments. Cytogenetic risk was evaluated according to Medical Research Council (MRC) criteria [25].

At diagnosis, *NPM1* and *FLT3*-ITD identification was performed by PCR amplification using gene-specific primers followed by capillary electrophoresis, according to Noguera et al. [26,27]. Patients were stratified based on the *FLT3*-ITD allelic ratio using the 0.5 cutoff, as defined in ELN 2017 guidelines [10,12]. Due to the limited cohort size, alternative thresholds were not tested statistically. The patients’ minimal residual disease (MRD) status was assessed before transplant through PCR amplification using NPM1 as a molecular marker, and in those without this molecular marker, MRD status was evaluated by multiparametric flow cytometry assay and/or WT1 expression [27,28].

### 2.2. Therapy and Disease Response Evaluation

Patients received “7+3” or “7+3 and midostaurin” induction chemotherapy. Drug regimens included cytarabine (100 mg/m^2^/day as a continuous intravenous infusion on days 1–7), daunorubicin (60 mg/m^2^/day intravenously on days 1–3) associated with midostaurin (50 mg twice daily orally on days 8–21). Midostaurin was administered only after its approval in Italy in 2019; therefore, treatment allocation was based on the treatment period rather than patient-specific criteria, resulting in a natural time-based cohort division.

Patients were reassessed between 21 and 28 days post-induction therapy. Treatment response was evaluated using the IWG 2003 criteria [29]. Complete remission (CR) was defined as the absence of leukemic blasts in the bone marrow (BM) (less than 5%), normal peripheral blood counts (neutrophils over 1000/μL and platelets over 100,000/μL), and no signs of extramedullary disease. Complete remission with incomplete count recovery (CRi) was similar to CR, but with either neutrophils or platelets still below normal levels. A Morphologic Leukemia-Free State (MLFS) was characterized by less than 5% blasts in the BM, but without full recovery of blood counts. Partial remission (PR) referred to a significant reduction in BM blasts, typically by at least 50%, or a reduction of less than 50% but with the percentage of blasts in the BM ranging from 5% to 20%, even though blood counts might not fully normalize. Those achieving CR proceeded to consolidation cycles, while patients with partial remission PR or refractory disease were removed from the study and monitored for survival outcomes. Consolidation therapy consisted of high-dose cytarabine (HiDAC), administered at 1000–3000 mg/m^2^ on days 1, 3, and 5, with or without midostaurin (50 mg twice daily from days 8–21).

Response rates were defined according to the 2010 or 2017 ELN criteria based on the date of diagnosis [12]. Patients at intermediate risk per the 2017 ELN guidelines or with positive minimal residual disease (MRD) in CR were referred for HSCT following local protocols. MRD status was assessed by WT1 expression or NPM1 quantitative real-time PCR (if *NPM1* mutation harbored).

### 2.3. Subgroup Analysis

Response rates (CR or CRi), DFS, and OS were analyzed across subgroups based on *FLT3*-ITD allelic ratio (AR ≥ 0.5 vs. AR < 0.5), *NPM1* mutation status (*NPM1*-mutated vs. non-mutated), midostaurin treatment (treated vs. untreated and analyzed for all patients and within low-AR and high-AR subgroups), and HSCT (bone marrow transplant in first CR vs. outside).

### 2.4. Statistical Analyses

Statistical analyses were performed using standard descriptive methods to summarize patient population characteristics and outcomes, including distribution measures, central tendency, and dispersion. In addition to descriptive statistics, time-to-event analyses were conducted using established methods, such as the χ^2^ test, Student’s *t*-test, Kaplan–Meier survival curves, and Cox proportional hazards models. Both univariate and multivariate analyses were employed to evaluate prognostic factors. Multivariable Cox proportional hazards models were constructed to assess the impact of prognostic variables on survival outcomes. A two-tailed *p*-value of <0.05 was considered statistically significant. All statistical analyses were performed using IBM SPSS v22©.3 (IBM, Armonk, NY, USA).

## 3. Results

### 3.1. Baseline Data

We evaluated 37 patients diagnosed with *FLT3*-ITD AML at our center between 2013 and 2022. Among them, 24 patients (64.8%) were categorized into the low-AR group (AR < 0.5), while 13 patients (35.2%) belonged to the high-AR group (AR ≥ 0.5), eight of them (61.5%) with AR ≥ 0.7 (i.e., RATIFY used cutoff). The mean age of all patients was 51.3 years (±9.1 SD), with no significant difference observed between the low-AR and high-AR groups (51.6 vs. 50.9, *p* = 0.83). Again, no differences were reported considering the genders. Regarding karyotype, one patient exhibited a low-risk karyotype with t(8;21) translocation, two patients had clonal aberrations conferring intermediate cytogenetic risk, 30 patients had a normal karyotype, one patient had a complex karyotype, and karyotype analysis failed for three patients, with no significant differences observed between the two AR groups (*p* = 0.39).

*NPM1* mutation was identified in 25 out of 37 patients (67.6%). In the low-AR group, this mutation was present in 16 out of 24 patients (66.7%), while in the high-AR group, it was present in 9 out of 13 patients (69.2%). Wild-type *NPM1* was detected in 12 out of 37 patients (32.4%).

Fifteen out of 37 patients (40%) underwent a “7+3” chemotherapy regimen from 2013 to 2018, while 22 out of 37 patients (60%) received “7+3” chemotherapy combined with midostaurin from 2019 to 2022. Among patients with low AR, 13 out of 24 (54.2%) underwent “7+3” chemotherapy alone, and 11 out of 24 (45.8%) underwent “7+3” chemotherapy combined with midostaurin. In contrast, among high-AR patients, 2 out of 13 (15.3%) were treated with “7+3” chemotherapy alone, while 11 out of 13 (84.6%) received “7+3” chemotherapy combined with midostaurin.

Baseline full blood count data and additional features are summarized in Table 1.

### 3.2. Induction Therapy and Overall Response

Overall response was achieved in 25 of 37 patients (67.6%). The response rates were similar between the low-AR group (16/24, 66.7%) and the high-AR group (9/13, 69.2%). Non-responders accounted for 32.4% of patients, with no significant difference between the groups (low-AR: 8/24, 33.3%; high-AR: 4/13, 30.7%; *p* > 0.99). When analyzing response based on *NPM1* mutation status, no statistically significant difference was observed overall. However, in the high-AR group, patients with *NPM1* mutation tended to a better response compared to those with wild-type *NPM1* (*p* = 0.05), as shown in Table 2.

When comparing the response rates between patients treated with “7+3” chemotherapy alone and those treated with a “7+3” scheme combined with midostaurin, which included 15 and 22 patients, respectively, no significant difference was observed in overall response rates between these groups (*p* = 0.49). Subgroup analyses were conducted for low-AR, high-AR, *NPM1*-mutated, and *NPM1* wild-type patients, finding no statistically significant differences (*p* = 0.2, 0.53, 0.17, and >0.99, respectively), suggesting a trend toward improved response in patients treated with “7+3” and midostaurin, particularly in the low-AR and *NPM1*-mutated subgroups, even if not statistically confirmed.

### 3.3. Measurable Residual Disease (MRD)

MRD status after induction therapy was assessed in 19 patients (51.3%), with negative MRD observed in 15 patients (78.9%; low-AR: 9/11, high-AR: 6/8) and positive MRD in 4 patients (21%; low-AR: 2/11, high-AR: 2/8). The two groups had no significant difference in MRD outcomes (*p* > 0.99).

### 3.4. Consolidation Therapy and HSCT

Consolidation therapy was administered to 24 of the 25 patients (96%) who achieved remission (1 patient was lost to follow-up after induction therapy). Among low-AR patients, the 16 (66.6%) that achieved CR underwent consolidation with HiDAC associated or not (before approval) with midostaurin, with an equal distribution of 1–2 and 3–4 cycles. In the high-AR group, the eight patients (61.5%) in CR received consolidation therapy, equally divided between 1–2 and 3–4 cycles. These data are summarized in Table 3.

HSCT was performed in 11 of 25 responder patients (44%). HSCT at first remission (CR1) was not a therapeutic choice in 14 patients due to their low ELN 2017 risk (10 cases) and early death (four cases) for septic shock and sudden aortic dissection. Considering patients in CR1, six patients had low AR, and five had high AR. Eight patients underwent HSCT after the second remission (CR2), with four (50%) in the low-AR group and four (50%) in the high-AR group.

Six out of 16 responders (37.5%) treated with midostaurin underwent HSCT, compared to five out of nine responders (55.6%) treated without midostaurin (*p* = 0.6).

### 3.5. Survival Outcome

At a median follow-up of 37.5 months (95% CI: 32.2–42.7), 26 patients died. DFS was evaluated in 25 responding patients. Median DFS was 8.6 months (95% CI: 3.7–13.5), as shown in Figure 1. DFS was significantly associated with *NPM1* mutation (*p* = 0.036) and HSCT at CR1 (*p* = 0.005) but not with AR or midostaurin treatment. In univariate analysis, when evaluating high-AR patients, the *NPM1* mutation, midostaurin treatment, and HSCT in CR1, significant associations were observed for *NPM1* mutation (HR: 0.320; 95% CI: 0.104–0.985; *p* = 0.047) and HSCT in CR1 (HR: 0.130; 95% CI: 0.025–0.695; *p* = 0.017).

Multivariate analysis, which included both *NPM1* mutation and HSCT in CR1, revealed that HSCT in CR1 remained statistically significant, with an HR of 0.160 (95% CI: 0.028–0.915; *p* = 0.039), while the association with *NPM1* mutation was not statistically significant.

The median OS for the entire cohort was 15.1 months (95% CI: 11.6–18.5), as shown in Figure 2. OS was significantly associated with HSCT but not with AR, *NPM1* mutation, or midostaurin treatment. OS comparison of midostaurin-treated patients versus non-treated patients in the low-AR group did not show any statistical difference (*p* = 0.09), such as when a similar comparison was made in the high-AR group (*p* = 0.5).

Significant findings include improved DFS with *NPM1* mutation (HR: 0.320, *p* = 0.047) and HSCT in CR1 (HR: 0.130, *p* = 0.017). OS showed no significant associations in multivariate analysis.

Focusing on patients who performed HSCT, 10 patients died due to transplant complication (three cases), or relapse (seven). The remaining 10 were alive at the cut-off data in CR.

Kaplan–Meier curves for DFS and OS stratified by allelic ratio, *NPM1* mutation, induction therapy, and HSCT are presented in Appendix A. Here, subgroup analyses, including those stratified by allelic ratio and *NPM1* mutation status, did not reveal statistically significant differences in response or survival outcomes associated with midostaurin use, suggesting a lack of clear benefit in specific subgroups, although small sample sizes limit definitive conclusions.

### 3.6. HSCT-Adjusted Outcome

When stratifying patients who achieved CR1, no significant survival benefit from HSCT was observed among patients in CR1 following non–midostaurin-based induction (Figure 3A). In contrast, those who underwent allogeneic HSCT following midostaurin-based induction showed significantly improved OS compared to those who did not undergo transplant (median OS not reached vs. 12.8 months; 95% CI, 6.9–18.7; *p* = 0.045, Figure 3B).

In a landmark analysis evaluating overall survival after allogeneic HSCT in CR1, patients who received midostaurin-based induction showed a trend toward improved OS compared to those who received non–midostaurin-based induction (median OS not reached vs. 11.5 months; 95% CI, 0.5–25.0; *p* = 0.086, Figure 4).

## 4. Discussion

AML with *FLT3* mutations remains a therapeutic challenge despite recent advances in targeted therapies and risk-adapted treatment strategies [13,17,30,31]. The response rates to induction therapy vary widely (40–80%), influenced by age, mutational profile, and treatment protocols [19,30,32]. In line with these data, our cohort achieved a response rate of 67.6%, with no significant difference between low and high-AR groups, nor between *NPM1*-mutated and wild-type subgroups.

The prognostic significance of the *FLT3*-ITD AR has long been a topic of discussion, and our findings appear to be in line with the 2022 ELN guidelines, which have progressively moved away from considering AR as an independent risk stratification criterion. While AR may still provide useful biological insight in specific contexts, its role as a standalone prognostic marker is increasingly being reconsidered in light of emerging evidence [13]. Although high AR has historically been linked to worse outcomes, recent data, including ours, suggest that the prognostic impact of *FLT3*-ITD may be modulated by concurrent mutations (e.g., *NPM1*), treatment strategy, and HSCT eligibility [13,33,34].

Our findings also reflect the historical use of the 0.5 allelic ratio cutoff, originally established based on its association with poor prognosis in earlier studies [10,12]. However, we acknowledge that this threshold has been increasingly debated. In our cohort, no alternative AR cutoffs were formally tested due to the limited sample size. While the pivotal RATIFY trial by Stone et al. adopted a 0.7 cutoff [17], we chose to use the 0.5 threshold in line with the ELN recommendations [12], and the real-world stratification routinely applied at our center. Moreover, the ELN group itself has validated the prognostic relevance of the 0.5 allelic ratio cutoff in midostaurin-treated patients, thereby reinforcing the findings of the RATIFY trial even when using the 0.5 threshold [10]. Given the observational and real-life nature of our study, this approach was considered most appropriate. Moreover, since midostaurin became available only after 2019, treatment allocation was based on the time of diagnosis rather than biological features. This non-randomized design may have introduced bias, which we attempted to adjust for using multivariable Cox regression models that included key prognostic variables such as *NPM1* mutation and HSCT.

Consistent with the RATIFY trial, we observed a trend toward improved outcomes with the addition of midostaurin to induction chemotherapy, particularly in low-AR and *NPM1*-mutated patients [17]. However, these improvements did not reach statistical significance in our cohort. Indeed, midostaurin demonstrated a benefit of midostaurin across all ELN risk categories, with 5-year OS rates of 73%, 52%, and 43% for favorable, intermediate, and adverse-risk groups, respectively, versus 53%, 34%, and 20% in the placebo arm [10]. Similarly, our real-world data suggest that midostaurin efficacy may be influenced by additional biological and clinical factors, as confirmed by other recent studies [35].

Our study presents several limitations that may have contributed to the lack of statistically significant findings regarding the impact of midostaurin. First, the relatively small sample size (n = 37), with only 22 patients receiving midostaurin, substantially reduces the statistical power to detect modest differences, such as the observed trend in response rates (72.6% vs. 60%, *p* = 0.49). Second, the cohort reflects a heterogeneous real-world population, in contrast to the more uniform criteria of the RATIFY trial, and includes patients treated over nine years (2013–2022). This temporal spread likely introduced variability in supportive care, MRD assessment, and transplant strategies. Third, the distribution of midostaurin exposure was unbalanced across subgroups: 85% of high-AR patients received midostaurin, compared to 46% of low-AR patients, which may have potentially confounded the interpretation of subgroup analyses. Moreover, MRD data were available for only half of the cohort, limiting our ability to assess remission depth and stratify relapse risk. Although we confirmed a significant benefit of HSCT in first remission (DFS HR 0.16, *p* = 0.039), transplant was performed in only 54% of cases, potentially attenuating the synergistic effects of midostaurin and post-remission consolidation. Lastly, while trends toward improved outcomes were observed in *NPM1*-mutated and low-AR patients treated with midostaurin, these did not reach statistical significance (e.g., *p* = 0.17 for *NPM1*-mutated), highlighting the need for larger, prospective studies to confirm these observations [36]. Notably, previous larger studies have also shown that the clinical benefit of midostaurin is more pronounced in NPM1-mutated patients, while its impact appears limited in those with *NPM1* wild-type AML [37]. This observation aligns with our findings and underscores the relevance of genetic context in modulating treatment efficacy. Another important limitation is the lack of standardization in measuring the *FLT3*-ITD AR across institutions, which may affect the reproducibility of results and complicate cross-study comparisons [10]. Moreover, the clinical utility of AR measurement is increasingly questionable in the post-midostaurin era, especially considering the stronger and more consistent prognostic value of co-mutations such as *NPM1* and of MRD assessment. While AR may still provide biological insights, our findings support the view that it should no longer be considered essential for clinical decision-making. Finally, the retrospective design and non-randomized allocation of midostaurin and HSCT may have introduced selection bias, which we attempted to mitigate through multivariable analysis, though residual confounding cannot be excluded

Secondarily, our study confirmed, already in a limited sized population, the critical role of allogeneic HSCT, particularly when performed in CR1. We observed significantly improved DFS in patients undergoing transplants. These findings align with prior studies indicating that HSCT improves both DFS and OS even in low-AR or favorable-risk patients [34]. From the RATIFY study, it emerged as well that patients who underwent HSCT at first remission had the best outcome in terms of post-transplant survival [17]. In the multivariate analysis from our dataset, HSCT in CR1 remained an independent factor for improved DFS (HR: 0.160, *p* = 0.039), reinforcing current guideline recommendations [13]. In addition, our real-life analysis revealed that patients who received midostaurin-based induction and subsequently underwent allogeneic HSCT in CR1 had significantly better OS compared to those not transplanted, whereas this benefit was not observed in patients induced without midostaurin. This finding is in line with the post hoc analysis from the RATIFY trial, which also suggested a trend toward improved post-transplant outcomes in patients treated with midostaurin during induction. In addition, the landmark analysis comparing post-HSCT in CR1 survival between midostaurin and non-midostaurin groups demonstrated a trend toward improved OS in the midostaurin cohort. These results collectively suggest that a sequential treatment with midostaurin-based induction followed by HSCT at CR1 can give a great survival advantage. These findings corroborate the potential synergistic benefit of combining midostaurin with allogeneic HSCT and support the incorporation of midostaurin in pre-transplant regimens for eligible *FLT3*-ITD AML patients.

MRD status, assessed in over half of the patients post-induction, was not significantly different between AR groups. However, MRD negativity was achieved in nearly 80% of tested patients, a finding associated with better outcomes in previous studies [38,39,40]. Yet, MRD assessment in *FLT3*-ITD AML remains complex. Due to clonal evolution and instability of *FLT3* mutations at relapse, *FLT3*-ITD MRD alone is not a reliable marker; a combined approach using NGS and PCR has been proposed to improve sensitivity [40]. Hopefully, this approach will pave the way for improved selection of patients for maintenance therapy post-HSCT. From the MORPHO trial, it emerged that patients with a peritransplant positivity for *FLT3*-specific MRD benefited more than others from post-transplant maintenance [41].

Another important consideration is the emergence of resistant clones during or after treatment with first-generation *FLT3* inhibitors such as midostaurin [1,42,43]. Second-generation *FLT3* inhibitors such as quizartinib and gilteritinib have shown more potent and sustained responses. For instance, the QuANTUM-First trial demonstrated significantly prolonged OS with quizartinib plus chemotherapy compared to placebo (31.9 vs. 15.1 months, *p* = 0.032) in newly diagnosed *FLT3*-ITD AML [44]. Recent comparative analyses have further contextualized the use of *FLT3* inhibitors in induction therapy. An anchored matching-adjusted comparison of the RATIFY and QuANTUM-First trials suggested that midostaurin and quizartinib offer broadly comparable OS benefits when combined with standard 7+3 induction chemotherapy in newly diagnosed *FLT3*-ITD AML, despite differences in study designs and patient populations [45]. In parallel, a large U.S. real-world retrospective cohort study reported similar treatment patterns and outcomes among patients treated with midostaurin and quizartinib [46].

Notably, both inhibitors demonstrated comparable response rates and early mortality, supporting the interchangeability of these agents in routine clinical practice, particularly in settings where prospective head-to-head comparisons are lacking. These findings emphasize the need for individualized treatment decisions based on patient-specific factors such as tolerability, co-mutations, and transplant eligibility.

## 5. Conclusions

Our findings align with the updated 2022 ELN classification, supporting the notion that *FLT3*-ITD AR should no longer be considered a decisive factor in routine prognostic assessment. In real-world practice, AR may hold limited clinical utility, especially in the presence of stronger prognosticators such as NPM1 mutation and MRD status. While NPM1 mutation was associated with better DFS, the most consistent survival benefit was observed with HSCT in first remission, reinforcing its central role in treatment strategy. The impact of midostaurin appears to be particularly pronounced when induction therapy is followed by consolidation with allogeneic HSCT. These findings support a shift away from AR-based stratification toward a more comprehensive approach that integrates target therapy-based induction, molecular characteristics, MRD assessment, and transplant-related factors to guide AML management.

## Figures and Tables

**Figure 1 jcm-14-05110-f001:**
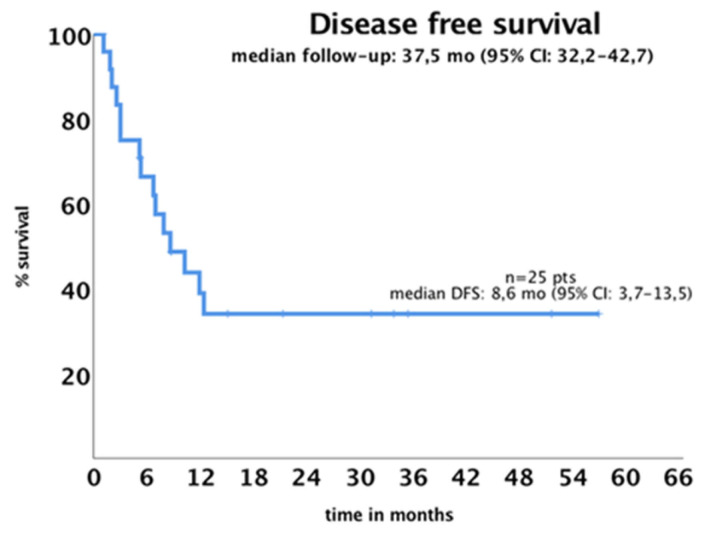
Disease-free survival for the 25 patients who achieved complete remission.

**Figure 2 jcm-14-05110-f002:**
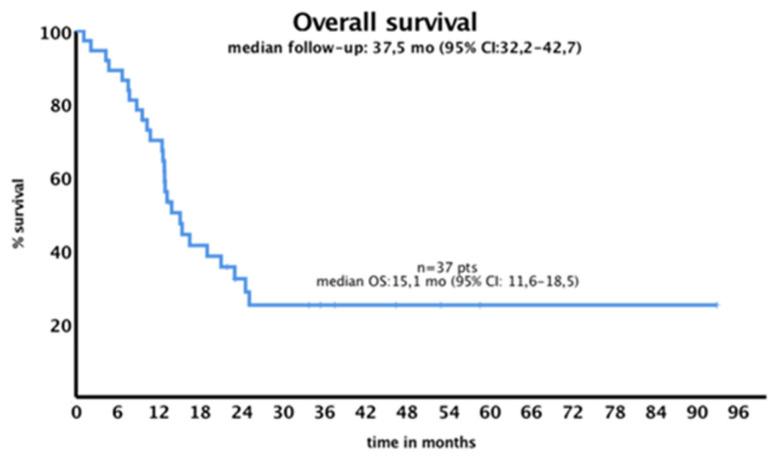
Overall survival for the whole cohort.

**Figure 3 jcm-14-05110-f003:**
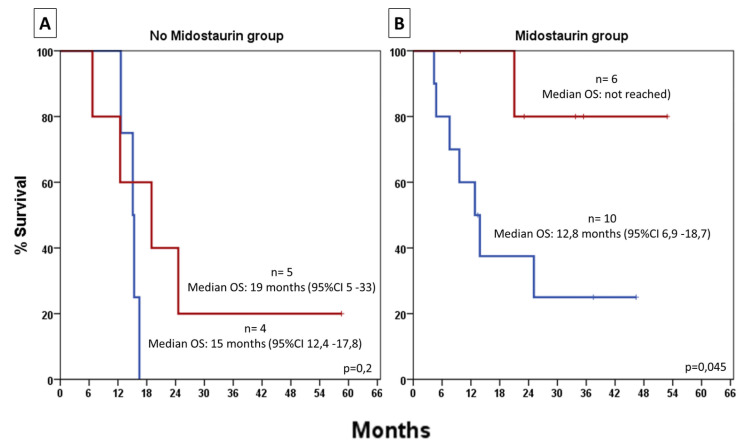
Overall survival (OS) of patients achieving first complete remission (CR1), stratified by whether hematopoietic stem cell transplantation (HSCT) was performed in CR1. Panel (**A**): patients who received induction with 7+3; Panel (**B**): patients who received 7+3 plus midostaurin; n = number of patients per group; shaded areas represent 95% confidence intervals (95% CI). Red line: patients who underwent HSCT in CR1; blue line: patients who did not undergo HSCT in CR1.

**Figure 4 jcm-14-05110-f004:**
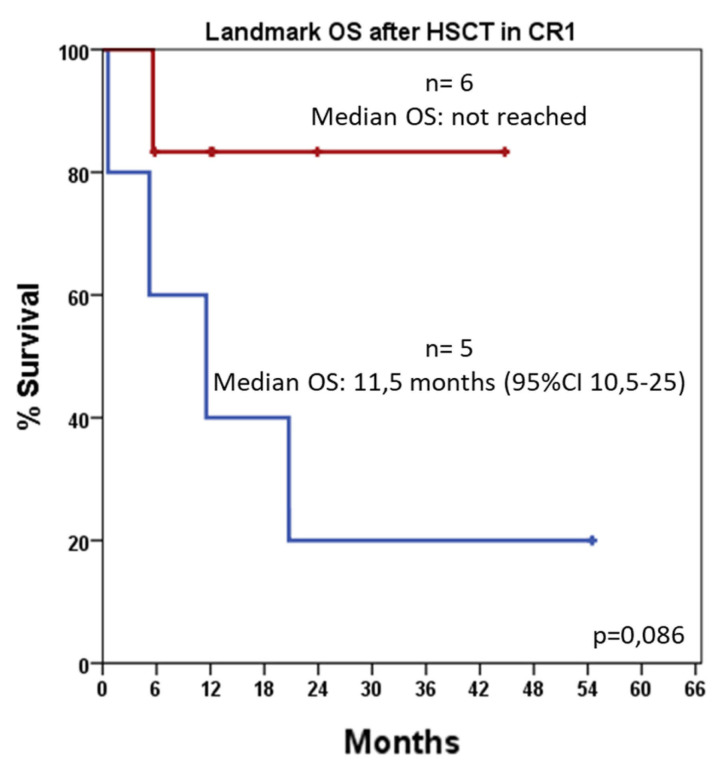
Landmark analysis of overall survival (OS) among patients who underwent hematopoietic stem cell transplantation (HSCT) in first complete remission (CR1), stratified by induction regimen: “7+3” (blue line) versus “7+3 plus midostaurin” (red line); n = number of patients per group.

**Table 1 jcm-14-05110-t001:** Baseline clinical data.

	37 Patients	Low-AR (n. 24)	High-AR (n. 13)	
Karyotype (3 failed)	1 low risk t(8;21) 2 clonal aberrations (intermediate) 30 normal 1 complex	18 normal 1 complex 1clonal aberration 1 low risk	12 normal 1 clonal aberration	*p* = 0.39
NPM1 mutated (%)	25/37 (67.6%)	16/24 (66.7%)	9/13 (69.2%)	*p* > 0.99
Average Hb g/dL (±SD)	8.7 (±1.8)	8.3 (±1.9)	9.4 (±1.5)	*p* = 0.07
Platelets/µL	91 × 10^3^ (±76 × 10^3^)	89 × 10^3^ (±59 × 10^3^)	94 × 10^3^ (±103 × 10^3^)	*p* = 0.83
WBC/µL	85 × 10^3^ (±117 × 10^3^)	71 × 10^3^ (±106 × 10^3^)	110 × 10^3^ (±138 × 10^3^)	*p* = 0.35
Chemotherapy “7+3” (yrs 2013–2018) (%)	15/37 (40%)	13/24 (54.2%)	2/13 (15.3%)	***p* = 0.035**
Chemotherapy “7+3” and midostaurin (yrs 2019–2022) (%)	22/37 (60%)	11/24 (45.8%)	11/13 (84.6%)	

Abbreviations: n.: number of patients; WBC: white blood cells; Hb: hemoglobin; SD: standard deviation; yrs: years; AR: allelic ratio.

**Table 2 jcm-14-05110-t002:** Response evaluation split according to AR, *NPM1* mutation.

**(A) Overall response (CR and CRi) and AR**			
	n. pts low-AR (24)	n. pts high-AR (13)	
Responders 25/37 (67.6%)	16/24 (66.7%)	9/13 (69.2%)	*p* > 0.99
Not responders 12/37 (32.4%)	8/24 (33.3%)	4/13 (30.7%)	
**(B) Overall response and NPM1 mutation**			
	*NPM1* mutated (25)	Wild type *NPM1* (12)	
Responders 25/37 (67.6%)	19/25 (76%)	6/12 (50%)	*p* = 0.14
Not responders 12/37 (32.4%)	6/25 (24%)	6/12 (50%)	
**(C) Overall response and NPM1 mutation in low-AR patients**			
	*NPM1* mutated in low-AR pts (16)	Wild-type *NPM1* in low-AR pts (8)	
Responders 16/24 (66.7%)	11/16 (68.7%)	5/8 (62.5%)	*p* > 0.99
Not responders 8/24 (33.3%)	5/16 (31.3%)	3/8 (37.5%)	
**(D) Overall response and NPM1 mutation in high-AR patients**			
	*NPM1* mutated in high AR (9)	Wild type *NPM1* in high AR (4)	
Responders 9/13 (69.2%)	8/9 (88.8%)	1/4 (25%)	*p* = 0.05
Not responders 4/13 (30.8%)	1/9 (11.2%)	3/4 (75%)	

Abbreviations: n. pts: number of patients; AR: allelic ratio; CR: complete remission; CRi: CR with incomplete hematological recovery; pts: patients.

**Table 3 jcm-14-05110-t003:** Number of cycles of consolidation split according to AR values.

N. pts/Total pts (%) 24/37 (64.9%)	N. pts/Total Low-AR (%) 16/24 (66.6%)	N. pts/Total High-AR (%) 8/13 (61.5%)
N. cycles of consolidation	1–2 cycles: 8 pts 3–4 cycles: 8 pts	1–2 cycles: 4 pts 3–4 cycles: 4 pts

## Data Availability

The data presented in this study are available on request from the corresponding author (the data are not publicly available due to privacy or ethical restrictions).

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
