# Peer review of "Real-World Outcomes in FLT3-ITD Mutated Acute Myeloid Leukemia: Impact of NPM1 Mutations and Allogeneic Transplantation in a Retrospective Unicentric Cohort"

_jcm, 2025, doi:10.3390/jcm14145110_

Round 1
Reviewer 1 Report
Comments and Suggestions for Authors
Duminuco et al. submitted a retrospective study evaluating the prognostic significance of FLT3-ITD allelic ratio (AR) in AML patients treated with intensive chemotherapy, with or without midostaurin, and explores its interplay with other molecular markers and clinical interventions like BMT. The study uses a sizable cohort of 149 patients and applies a variety of statistical tools. The work is timely, particularly considering evolving treatment paradigms for AML. However, significant clarifications are needed, so I have some suggestions/comments, written section-by-section, that might be taken into consideration prior to the publication.
1. Title: It is better to clarify that the study is a retrospective cohort study in the title.
2. Abstract: I recommend that authors explicitly state whether midostaurin improved outcomes in high-AR vs. low-AR groups in terms of both OS and DFS and clarify the conclusion to emphasize clinical implications more definitively.
3. Introduction: I would suggest expanding on previous conflicting findings from RATIFY and real-world datasets and on the rationale behind the historical AR cutoff of 0.5. In addition, authors should clarify whether current guidelines still advocate for risk stratification based on AR in light of this study's implications.
4. Methods: It is very important to clarify inclusion/exclusion criteria, especially regarding BMT eligibility and specify how patients were grouped into high-AR and low-AR and the rationale behind the chosen threshold. I would also like to know how the rationale for using 0.5 as the AR cutoff and whether alternative thresholds were tested. Also, I have a concern whether a multivariable Cox model was performed to adjust for known prognostic variables. In addition, authors should specify whether therapy allocation (e.g., midostaurin use) was random or based on time periods or patient characteristics.
5. Results: Please provide hazard ratios and 95% CIs for survival curves and avoid overstating clinical relevance when statistical difference is not reached (e.g., midostaurin impact on high-AR patients). Could you please consider tabulating BMT-adjusted outcomes separately? It is important to clarify if all patients treated with midostaurin also received similar post-remission therapy (e.g., transplant). In supplementary results, many subgroup comparisons (e.g., midostaurin use in AR-defined groups) show non-significance; consider summarizing these with a clearer interpretation of lack of benefit.
6. Discussion: It would be better to mention the lack of standardization in AR measurement across centers as a limitation. Authors should expand on the clinical utility of measuring AR given the findings—should we still measure it? Also, authors should discuss other markers (e.g., MRD, co-mutations like NPM1) as potentially stronger prognosticators in the post-midostaurin era. Acknowledge potential bias due to retrospective data and selection of patients for midostaurin or BMT.
7. Conclusion: I would like to see a clear statement on the clinical implications, such as whether FLT3-ITD AR should continue to guide treatment decisions in AML in real-world practice and reinforce the message for practicing clinicians; what might change in routine prognostic assessment?
P.s. Plagiarism percentage (26%) is quite high, please reduce it.
Reviewer 2 Report
Comments and Suggestions for Authors
General Comments:
- Relevance and Novelty: The manuscript addresses an important clinical issue by assessing real-world outcomes in FLT3-ITD mutated AML, especially regarding NPM1 mutations and HSCT. The topic is timely and contributes valuable real-world data, particularly in light of the 2022 ELN guideline changes.
- Clarity and Organization: The manuscript is well-organized, and the flow from background to conclusions is logical. Figures and tables are useful and help support the narrative. However, some typographical issues should be revised.
- Sample Size and Statistical Power: The limited sample size (n=37) and especially the subgroup divisions reduce the power to detect significant differences. This limitation is acknowledged in the discussion but should be explicitly highlighted earlier in the methods section.
Specific Comments:
Introduction:
- The background on AR and its evolving role is well-presented. It might help to briefly mention why real-world data may differ from clinical trial outcomes.
Methods:
- The rationale behind choosing 2013–2022 as the study window could be explained.
- Clarify if midostaurin use was strictly post-2018 (when approved in Italy) or if exceptions occurred.
Results:
- Tables are informative, but Table 1 is visually dense. Consider separating demographic/clinical characteristics into two sub-tables.
Discussion:
- You effectively relate findings to the 2022 ELN classification.
- The limitations section is robust and appropriately critical. However, the potential biases introduced by treatment era differences (pre- and post-midostaurin approval) deserve more emphasis.
Conclusion:
- The conclusion is appropriate, but the final sentence could be stronger. For example:
"Larger, prospective multicenter studies are needed to validate these findings and optimize treatment pathways in FLT3-ITD AML."
Language and Style Suggestions:
- Minor typographical issue are scattered throughout. Examples: "midostaurine” should be "midostaurin"
Final Recommendation:
Minor revision. The manuscript addresses a clinically relevant question with useful real-world insights. With refinement in language and clarification of methodological details, the manuscript would be suitable for publication.
Round 2
Reviewer 1 Report
Comments and Suggestions for Authors
All comments have addressed properly.